# Genome Wide Analysis of Family-1 UDP Glycosyltransferases in *Populus trichocarpa* Specifies Abiotic Stress Responsive Glycosylation Mechanisms

**DOI:** 10.3390/genes13091640

**Published:** 2022-09-13

**Authors:** Hafiz Mamoon Rehman, Uzair Muhammad Khan, Sehar Nawaz, Fozia Saleem, Nisar Ahmed, Iqrar Ahmad Rana, Rana Muhammad Atif, Nabeel Shaheen, Hyojin Seo

**Affiliations:** 1Centre for Agricultural Biochemistry and Biotechnology, University of Agriculture Faisalabad Pakistan, Punjab 38000, Pakistan; 2Center for Advanced Studies in Agriculture and Food Security, University of Agriculture Faisalabad Pakistan, Punjab 38000, Pakistan; 3Department of Plant Breeding and Genetics, University of Agriculture Faisalabad Pakistan, Punjab 38000, Pakistan; 4Korea Soybean Research Institute, Jinju 52840, Korea

**Keywords:** *Populus trichocarpa*, plant secondary product glycosyltransferase, drought, genome-wide, co-expression, abiotic stresses

## Abstract

*Populus trichocarpa* (Black cottonwood) is a dominant timber-yielding tree that has become a notable model plant for genome-level insights in forest trees. The efficient transport and solubility of various glycoside-associated compounds is linked to Family-1 UDP-glycosyltransferase (EC 2.4.1.x; UGTs) enzymes. These glycosyltransferase enzymes play a vital role in diverse plant functions, such as regulation of hormonal homeostasis, growth and development (seed, flower, fiber, root, etc.), xenobiotic detoxification, stress response (salt, drought, and oxidative), and biosynthesis of secondary metabolites. Here, we report a genome-wide analysis of the *P. trichocarpa* genome that identified 191 putative UGTs distributed across all chromosomes (with the exception of chromosome 20) based on 44 conserved plant secondary product glycosyltransferase (PSPG) motif amino acid sequences. Phylogenetic analysis of the 191 *Populus* UGTs together with 22 referenced UGTs from *Arabidopsis* and maize clustered the putative UGTs into 16 major groups (A–P). Whole-genome duplication events were the dominant pattern of duplication among UGTs in *Populus*. A well-conserved intron insertion was detected in most intron-containing UGTs across eight examined eudicots, including *Populus*. Most of the UGT genes were found preferentially expressed in leaf and root tissues in general. The regulation of putative UGT expression in response to drought, salt and heat stress was observed based on microarray and available RNA sequencing datasets. Up- and down-regulated UGT expression models were designed, based on transcripts per kilobase million values, confirmed their maximally varied expression under drought, salt and heat stresses. Co-expression networking of putative UGTs indicated their maximum co-expression with cytochrome P450 genes involved in triterpenoid biosynthesis. Our results provide an important resource for the identification of functional UGT genes to manipulate abiotic stress responsive glycosylation in *Populus*.

## 1. Introduction

The global ecological and economic importance of a terrestrial ecosystem is contingent upon the number of forest trees present in that geographical area [1,2]. *Populus* is a model species for perennial woody plants and a valuable forest resource for timber yield worldwide. It is a model woody plant because its genome resources (including cDNA clones, expressed sequence tags, microarray data sets and RNA seq data sets) are readily available [3,4,5]. Additionally, it is known as a potential candidate tree for carbon sequestration, nutrient cycling, and phytoremediation [1,6]. *Populus* trees have evolved a diverse set of defense systems to cope with a diverse range of biotic and abiotic stresses over their evolution [1].

Glycosylation is one of the final steps involved in the triterpenoid biosynthesis pathway for many plant defensive compounds, such as phenolics, glucosinolates, salicylates, and anthocyanins [7,8,9,10]. Glycosyltransferase (GT) family 1 has been found to be the largest gene family in the plant kingdom [9,11]. GTs can transfer sugar moieties from active sugar molecules to a variety of acceptor molecules, and are hence referred to as uridine diphosphate glycosyltransferases (UGTs) [7]. These enzymes have a 44-amino acid consensus sequence near the C terminal, referred to as the plant secondary product glycosyltransferase (PSPG) box [12,13,14,15,16]. Using this PSPG motif as a search tool, 107 putative UGT genes were identified in the *Arabidopsis thaliana* genome [17]. Subsequently, putative UGT genes have been identified in other plants, including *Prunus persica* (168 genes), *Malus domestica* (254), *Vitis vinifera* (184), *Linum usitatissimum* (138), *Glycine max* (148), *Zea mays* (148), *Glycine soja* (128), *Gossypium hirsutum* (274), *Eucommia ulmoides* (91), Apple (237), *Quercus robur* (244), *Vitis vinifera* (257), *Oryza sativa* (41) *Camellia sinensis* and *Triticum aestivum* (179) [12,18,19,20,21,22,23,24,25,26,27,28,29]. A 40% amino acid similarity can be found among the UGTs, whereas 60% or greater similarity has been observed within subfamilies [13,30,31,32]. On the basis of phylogeny, UGTs can be divided into 16 distinct groups (A–P). In *Zea mays*, 17 distinct phylogenetic groups (A–Q) have been observed [21]. To date, 115 GT families have been identified in the CAZy (carbohydrate-active enzyme) database (http://www.cazy.org/ accessed on 3 February 2022) [33], which is a comprehensive resource that specializes in organizing carbohydrate-active enzymes [34,35]. GTs comprise almost 40% of the enzymes present on the CAZy website [34].

UGTs respond to a variety of plant stresses by conjugating with various phytohormones [36]. However, their biological roles in response to abiotic stresses are largely unknown. For example, in *Arabidopsis*, *UGT85U1/2* and *UGT85V1* were found to be involved in salt and oxidative stress tolerance by changing the composition of several indole derivatives [37]. In tobacco, ectopic expression of UGT85A5 resulted in enhanced salt stress tolerance in transgenic plants [38]. Further, in *Arabidopsis*, ectopic over-expression of *UGT74E2* increased the tolerance to salinity and drought stress and reduced the plants’ water loss [39], and 11 UGTs were up-regulated by H_2_O_2_ stress in catalase-deficient plants [40]. Moreover, *GT85C2*, *UGT74G1,* and *UGT76G1* were shown to be down-regulated under polyethylene glycol-induced drought stress in *Stevia rebaudiana* [41]. The involvement of abscisic acid (ABA) in mediating drought stress has been extensively researched and several UGTs have been functionally characterized for ABA-glucose ester formation [42,43,44,45]. ABA glycoconjugate genes from various plants have been characterized, including *UGT71* (Strawberry); *UGT87A2*, *UGT71B7*, *UGT71B8*, *UGT71B6*, *UGT75B1* and *UGT71C5* (*Arabidopsis*); *ABAGT* (*Phaseolus vulgaris*); *UGT73C14* (*Gossypium hirsutum*); and *ABAGT* (*Vigna angularis*) [42,43,44,45,46,47,48,49]. In contrast to their function in response to abiotic stresses, the role of UGTs in response to biotic stresses has been well characterized. For example, in tomato (*UGT73B3*, *UGT73B*) and *Arabidopsis* (*AtSGT1*, *UGT76B1*,) genes have been characterized in response to *Pseudomonas syringae* infection, and possibly play a role in salicylic acid and jasmonic acid crosstalk [50,51,52,53].The role of UGTs in fiber and seed development, specifically, has been examined using RNA-seq data in cotton, flax, and soybean [18,20,23].

Here, we employed bioinformatics techniques to identify 191 putative UGTs in *Populus*, and then subjected these to genomic and expression analyses. We then further confirmed the publicly available microarray and transcriptomic data on UGT expression in plant tissues, drought, salt, cold and heat stresses. We designed UGT expression models for up- and down-regulation of UGT genes under drought and control conditions. Finally, co-expression networks were constructed in order to gain a better understanding of the involvement of these genes in other pathways.

## 2. Materials and Methods

### 2.1. Identification of Putative UGTs in the Populus Genome

The *Populus* genome was searched for putative UGTs through the popgenie database (http://popgenie.org/ accessed on 1 February 2022). Forty previously characterized UGTs were used as models to produce an HMM model of the typical PSPG-box found in UGTs [5]. In addition, 19 reference UGT protein sequences were obtained from the *A. thaliana* cytochrome P450 website (http://www.p450.kvl.dk accessed on 1 February 2022) and a further 3 from the *Z. mays* genome database (https://www.maizegdb.org/ accessed on 2 February 2022) for subsequent phylogenetic tree construction (Appendix A). To retrieve all the putative UGT sequences from the *Populus* genome, BLAST (BLASTP) 2.2.28 was performed on the popgenie website using the PSPG motif from each phylogenetic group published by [12] as the query sequence. The following search parameters for BLASTP were used: scoring matrix (BLOSUM62) and expectation value (1e-10). The consensus pattern of the designed HMM model [FW]-x(2)-[QL]-x(2)-[LIVMYA]-[LIMV]-x(4,6)-[LVGAC]-[LVFYAHM]-[LIVMF]-[STAGCM]-[HNQ]-[STAGC]-G-x(2)-[STAG]-x(3)-[STAGL]-[LIVMFA]-x(4,5)-[PQR]-[LIVMTA]-x(3)-[PA]-x(2,3)-[DES]-[QEHNR] was further used to search for this pattern in all retrieved sequences by using the FUZZPRO program (http://www.bioinformatics.nl/cgi-bin/emboss/fuzzpro accessed on 4 February 2022). Graphical representations of the PSPG motif in putative UGTs were also obtained using the ScanProsite website (http://prosite.expasy.org/prosite.html accessed on 5 February 2022). Functional annotations, such as Alias and GO ontology, chromosomal location, peptide length, PFAM id, pathway information and *Arabidopsis* synonyms of each UGT were obtained from the popgenie and Phytozome databases.

### 2.2. PCR Cloning and Sequencing of Selected Populus UGTs

To validate the UGT sequences obtained, we extracted genomic DNA from *P. trichocarpa* leaves using a DNeasy Plant Mini Kit (Qiagen GmbH, Germany). The extracted DNA quality and quantity were measured using a NanoDrop 1000 spectrophotometer (Thermo Scientific Inc., USA). PCR amplification was performed using 30 ng of genomic DNA as a template, with TAKARA Ex Taq HS polymerase (Clonetech, Japan). The UGT genes *Potri.016G097400*, *Potri.006G120600*, *Potri.010G182575*, *Potri.016G057300*, and *Potri.016G057300* were selected, and their full reading frame was amplified using the primers and annealing temperature shown in Appendix A. Amplicons were visualized on 0.8% agarose gels and eluted using an AccuPrep Gel Purification Kit (Bioneer Inc., Daejeon, Korea). Eluted amplicons were cloned into pGEM-T Easy vectors (Promega, USA) and sequenced using the BigDyeTM terminator method in an ABI 3730XL DNA analyzer (Bioneer Inc.).

### 2.3. Motif Alignment, Phylogenetic Analysis, and Comparison

Downloaded sequences were aligned with MUSCLE using MEGA 7 software [54]. Sequences that were too short, too divergent, or too long were removed from the input file after initial alignment and were re-aligned. The obtained alignment file contained only those sequences similar to the desired PSPG motif. Phylogenetic analysis was performed using the Neighbor-joining statistical method with 1000 bootstrap replications in MEGA 7.0.1.18. A 100% data coverage was used to construct the phylogeny. The phylogenetic tree was visualized using the iTOL website (http://itol.embl.de/ accessed on 5 February 2022). To compare the phylogenetic groups of putative UGTs in *Populus*, we collected published data on the number of putative UGTs and the number of phylogenic groups including *Prunus persica*, *Malus domestica*, *Vitis vinifera*, *Linum usitatissimum*, *G. max*, *Glycine soja*, *G. hirsutum*, *Gossypium raimondii* and *Gossypium arboreum* [12,18,19,20,21,22,23,24,25,26,27,28].

### 2.4. Gene Duplication and Chromosomal Distribution

The physical location of each UGT on chromosomes was retrieved using the start and stop positions of genes taken from the Phytozome database (https://phytozome.jgi.doe.gov/pz/portal.html accessed on 6 February 2022). Mapchart 2.2 was used to visualize UGT gene distribution on *Populus* chromosomes [55]. A gene cluster was defined as two or more copies located in a chromosomal region <200 kb [56].

To predict UGT duplication patterns, we examined segmental and tandem duplication types. Tandem duplications were defined based on more than one gene family member located in the same or neighboring regions of the genome. If the two *UGTs* were located on duplicated blocks and highly similar at the amino acid level, they were considered segmental duplication events. The Plant Genome Duplication Database (PGDD) server (http://chibba.agtec.uga.edu/duplication/ accessed on 7 February 2022) was used to retrieve duplications for each UGT gene. Within a 100-kb range, anchors with identity >1.0 were rejected to avoid saturation [57]. Assuming the operation of a molecular clock, the synonymous substitution (Ks) values of duplicated genes are probably similar over time [58]. Hence, to estimate the dates of segmental duplication events, Ks values were cast off and their means calculated for each gene pair inside a duplicated block. The estimated date of the duplication event was then predicted using the mean Ks values (T = Ks/2λ), assuming clock-like rates (λ) of 9.1 × 10^−9^ identical substitutions/synonymous site/year for *Populus* [59].

Gene duplications were further confirmed by searching for all branching points in the topology with at least one species that is present in both subtrees of the branching point. An unrooted gene tree was used for the analysis, such that the search for duplication events was performed by identifying the placement of the root on a branch or branches that produced the minimum number of duplication events. Again, MEGA7 was used to determine the evolutionary relationships of taxa and a duplication tree was generated. Finally, a syntenic analysis among all the retrieved sequences was performed using the Circoletto webtool (http://tools.bat.infspire.org/circoletto/ accessed on 10 February 2022) by providing the FASTA sequences. The resulting image was captured.

### 2.5. Gene Structure Analysis

The exon/intron organization for each phylogenetic group was illustrated using the Gene structure display server (GSDS) program (http://gsds.cbi.pku.edu.cn/ accessed on 10 February 2022), by aligning the coding and genomic sequences obtained from Phytozome. Introns were classified based on the structure in the genome, including phase, length, and number. A UGT intron map was constructed in accordance with a previously established method [18,23]. The insertion events were serially numbered as I-1 to I-10, according to their positions [18,25]. The splice sites of intron-containing UGTs were mapped onto all aligned sequences of intron-containing UGT peptides using the PIECE web tool (https://wheat.pw.usda.gov/piece/ accessed on 11 February 2022). The gene structure of putative UGTs was also compared with other UGTs from *Arabidopsis*, *P. persica*, *M. domestica*, *V. vinifera*, *L. usitatissimum*, *Glycine max*, and *Gossypium raimondii* using the PIECE webtool. The gene structure was compared based on PFAM domain PF00201.

### 2.6. Digital Expression Analysis

Microarray data of the putative UGTs in different tissues, such as leaves, phloem/cambium, roots, petioles, twigs, buds, flowers, and suckers, were obtained from the popgenie database using the Asp201 EA Affymetrix dataset. A heatmap of the data was generated with distance function (Euclidean) and hierarchical clustering (Average), using the exHeatmap tool on the popgenie website. The same Affymetrix dataset was used to check the expression of these UGT genes under different conditions, such as drought, heat, cold and salt stresses. A heatmap was generated using the previously mentioned conditions.

### 2.7. RNA Sequencing based UGTs Model in Tissues and under Drought Stress

Digital expression of all UGTs related to their maximum varied expression under drought conditions was further evaluated using RNA sequence data taken from [60]. The 24,013 transcripts were analyzed from non-stressed (control) and drought-stressed leaves of *Populus euphratica*, and we retained those with transcripts per kilobase million (TPM) values of one to maximum in both control and drought-stressed samples [60]. On average, a 1128-bp length was recorded for the yielded transcripts after assembly [60]. Normalized TPM values were calculated for all expressed UGTs under control and drought conditions. The TPM values of each expressed UGT under control and drought were tabulated according to their phylogenetic group and a final graph was drawn for the up-regulated and down-regulated UGTs under drought.

Furthermore, RNA seq data values for tissues (leaf, root, shoot, xylem, phloem, fiber and vessel) and stresses were sourced from Plant FIBer Expression database https://ssl.cres-t.org/cgi-fibex/cluster0.cgi?sp=lu_pt (accessed on 12 February 2022) and then were visualized in the Morphus heat map tool.

### 2.8. Co-Expression Networking of Putative UGTs

Gene and protein interaction networks were examined to detect associations between putative UGTs and biological processes using the famNet database (http://aranet.mpimp-golm.mpg.de/famnet.html accessed on 14 February 2022) tool, which generated co-expression networks for each UGT gene. The gene identifier of each UGT was searched for in the famNet database, which enabled us to generate images of the resulting networks. The networks contained all nodes supported by ELA (ensemble label association) and all genes one step away from the query gene were drawn [61]. We considered gene networks of a single query gene having maximum co-occurrences in order to enhance our understanding of the gene families co-expressed with putative UGTs in *Populus*. An overall co-expression image was drawn by comparing the individual gene networks.

## 3. Results 

### 3.1. Identified Putative UGTs in the Populus Genome

We identified 191 putative UGTs possessing the C-terminal PSPG motif from the projected soybean proteome (Appendix A and Figure 1A, B). The tools used for identification of UGTs produced consistent results in terms of PSPG motif presence. The lengths of the deduced UGT proteins ranged from 511 to 117 amino acids, with an average length of 465. All UGT sequences started with a methionine and were full-length. The 191 putative UGTs were found to be involved in nine different biosynthesis pathways, namely, saponin biosynthesis, scopolin and esculin biosynthesis, daphnetin modification, myricetin gentiobioside biosynthesis, rutin biosynthesis, cytokinin-glucoside biosynthesis, anthocyanin biosynthesis, and dhurrin biosynthesis (Appendix A). Go ontology analysis showed that most of the UTGs are involved in transferring hexosyl groups and metabolism.

### 3.2. Motif Alignment, Phylogenetic Analysis, and Comparison

Cloned UGTs from chromosomes 16, 10, and 6 have sequences similar to those obtained from the popgenie database. The final alignment file contained 503 aligned positions with two highly conserved, 494 variable, and 20 singleton sites. The PSPG motif was found to vary in each phylogenetic group (Appendix A and Figure 1B). The histidines at positions 10 and 19 in the PSPG motif were the most conserved.

The constructed phylogenetic tree indicated a classification of 16 major groups (A–P), including two newly discovered groups (O and P), which are absent in *Arabidopsis* but present in higher plants like maize, cotton, peach, and apple (Figure 1A) [18,20,21,25]. All the known phylogenetic groups were present in *Populus* except Q, which is present only in *Zea mays* [21].

Overall, among all eudicots, groups A, D, G, E, and L have the largest numbers of UGTs (Figure 2). Phylogenetic group E in *Populus* was the most expanded one among all the compared eudicots except *M. domestica*, which had 66 genes (Figure 2). Group N was highly conserved among all eudicots, each having one gene (Figure 2). The newly identified groups O and P each contained three genes (Figure 2).

### 3.3. Gene Duplication and Chromosomal Distribution

With the exception of chromosome 20, all *Populus* chromosomes contained UGTs (Figure 3), although the numbers varied. Chromosomes 16, 06, 01, and 17 possessed the highest numbers of UGTs (44, 24, 19, and 17, respectively), whereas chromosomes 03, 11, and 13 had only three UGTs. A total of 49 clusters were found, with sizes ranging from 2 to 15 genes per cluster (Figure 3). Chromosomes 16, 06, 09, and 17 had the largest clusters. Maximum clustering was found in groups E and G. Twenty-two UGTs did not cluster.

Sixty-five of the total 191 UTGs exhibited segmental duplication (Appendix A). Of these, 44 have single segmentally duplicated loci and 21 have two to four loci. Interestingly, two UGTs from group G, *Potri.002G098300* and *Potri.002G098300*, were duplicated at four different loci. Moreover, three tandem duplications were also observed. Phylogenetic groups G, E, D, and L contain the largest number of segmentally duplicated UGTs (19, 12, 9, and 5, respectively), whereas both O and P have only four. Group I is the only phylogenetic group having three tandemly duplicated UGTs. All phylogenetic groups contain duplicated genes but the numbers vary.

The estimated time for segmental duplication in the 65 segmentally duplicated UGTs ranged from 11 to 189 mya. Approximately 40 UGTs were segmentally duplicated at around 11–65 mya, whereas duplication of the remaining 25 UGTs occurred during the period 65–189 mya (Appendix A). We also calculated the time of duplication within each group (Appendix A). For some UGTs, there were two different duplicated loci, duplicated at different times on different chromosomes. For example, *Potri.002G168600* was segmentally duplicated from *Potri.003G138200* and *Potri.014G095900* at around 107 and 138 mya. Some UGT genes have recently undergone segmental duplication because they have no Ka/Ks values in the duplication database (Appendix A). Thus, *Populus* UGT evolution involves a series of segmental duplications, during which members of groups E, M, C, D, L, I, M, and F have been duplicated twice, whereas members of groups A and G have been duplicated two, three, or four times. Whole-genome duplication (WGD) and syntenic analysis using MEGA7 and Circoletto confirmed that all 191 UGTs were duplicated from each other (Figure 4 and Appendix A).

### 3.4. Gene Structure Analysis

Of the 191 *UGT*s, 83 possessed no introns, whereas 92, 12, one, and three possessed one, two, three, and four introns, respectively (Appendix A and Appendix A). A maximum of four introns were found in the following UGT genes: *Potri.014G082500* (*UGT73*), *Potri.001G303100* (*UGT73*), and *Potri.007G140600* (*UGT74*). A total of 133 introns were found across all intron-containing UGTs. Among these, 7 introns were present in phase 2, 35 in phase 1, and 31 in phase 0 (Appendix A and Appendix A). A total of 77 introns were found in phase 1, 27 in phase 0, and 29 in phase 2 (Figure 5). Collectively, intron sizes ranged from 6 to 6299 bp. Of the 68 introns, nine were longer than 2000 bp Appendix A). An average of 1.23 introns/gene was found among all intron-containing UGTs. The intron size ranged from 1 to 8308 bp in length among all the intron-containing UGTs. Intron size was also compared between the phylogenetic groups: A (1–4826), D (2–6320 bp), E (5–4649 bp), F (161–1832 bp), G (4–8308 bp), I (124–255 bp), L (2–4555 bp), M (50–4638 bp), K (25–284 bp), H (103–5417), and P (214–434 bp). Group B, C, and O UTGs did not have introns. Intron phases in some phylogenetic groups were also found to be conserved; for example, all 37 introns in group G UTGs were found in phase 1.

After mapping the introns to the amino acid sequence alignments, at least 10 independent intron insertion events were observed by following [18,25]. These insertion events are serially numbered I-1 to I-10 according to their positions (Figure 5). Insertion between 150–200 aligned amino acids is highly conserved in phase 1. For group G, 35 out of 36 UGTs have highly conserved introns between these amino acids insertions. Another insertion in intron was predominantly observed only in group L. Interestingly, all the conserved intron insertions were found only in phase 1. Seventeen UGTs out of 53 from group E have introns with deletion of the conserved insertion.

We identified 1632 UGT transcripts in *Arabidopsis*, *P. persica*, *M. domestica*, *V. vinifera*, *L. usitatissimum*, *G. max*, *G. raimondii,* and *Populus,* based on a comparison of gene structure and conservation of introns using the search keyword PFAM00201, on the PIECE website. Six hundred and thirty-two transcripts were found to contain no introns. All the remaining 1010 intron-containing UGTs from these eudicots have the same conserved intron and phase 1 (Appendix A).

### 3.5. Digital Expression Analysis

Microarray data obtained from the popgenie database identified 164 UGTs with varied expression across roots, leaves, flowers, buds, phloem/cambium, twigs, seeds, and suckers (Appendix A). Most of the UGTs were abundantly expressed in flowers, roots, and leaves. Least expression was found in phloem/cambium tissues. A range of UTG expression levels was also found with respect to the following conditions: drought, mechanical damaged, beetle damage, dormancy, girdled and non-girdled, whole sucker, expanding and expanded, young and freshly expanded (Figure 6). On the basis of this microarray data, there is a clear up- and down-regulation of these UGTs genes in response to drought conditions (Figure 6). Genes preferentially expressed in response to drought were observed from the following phylogenetic groups: *Potri.001G030600* (*UGT91*) from group A, *Potri.007G030300* (*UGT72*) and *Potri.012G036000* (*UGT88*) from group E, *Potri.001G303300* (UGT73) and *Potri.009G099000* (*UGT73*) from group D, *Potri.001G281800* (*UGT87*) and *Potri.009G077500* (*UGT87*) from group J, *Potri.009G039000* (*UGT76*), *Potri.004G119700* (*UGT83*), and *Potri.006G022500* (*UGT85*) (Figure 6). Genes from groups B, C, M, and F were also expressed but at low levels.

### 3.6. RNA-Seq-based UGT Model under Drought

On the basis of RNA-seq data, we identified 152 UGTs differentially expressed under control and drought stress condition in *Populus*
*euphratica* leaves (Appendix A). Eighty-four UGT genes were found to be either up- or down-regulated under control and drought conditions based on their TPM values (Figure 7A,B). In total, we identified 42 UGT genes that were down-regulated under drought stress, belonging to the following phylogenetic groups: E, M, B, C, D, L, K, and F. Groups E and L contained the highest numbers of down-regulated UGTs—20 and 13, respectively (Figure 7A). Groups, B, C, M, D, L, and F share only nine UGT genes in common. *Potri.016G014500* (*UGT71*) from group E was maximally down-regulated under drought conditions, with 23.32 TPM compared to the control with 218.76 TPM. *Potri.009G095100* (*UGT84*) from group L was down-regulated under drought conditions, with a TPM value of 76.42 compared to the control with a TPM value of 137.38. Groups A, P, and O contained no UGT genes that were down-regulated under conditions of drought stress (Figure 7A and Figure 8).

The same number of genes (42) was found to be up-regulated under drought conditions, and these belong to the following phylogenetic groups: E, A, M, D, O, L, J, H, and P (Figure 7B). The highest differential expression under drought conditions was observed in the UTGs from groups L, E, and P. For example, *Potri.017G032300* (*UGT74*), *Potri.017G032500* (*UGT74*), *Potri.017G032700* (*UGT74*), *Potri.001G389200* (*UGT74*), *Potri.007G140500* (*UGT74*), and *Potri.007G117200* (*UGT74*) from group L were highly up-regulated under drought stress. The group E UGTs *Potri.009G044600* (*UGT71*) and *Potri.006G010000* (*UGT71*) were also found to be up-regulated under drought stress. Interestingly, one UGT, *Potri.016G105400* (*UGT70*), from the newly formed phylogenetic group P was also found to be up-regulated, with a TPM value of 61.25, under drought stress compared with 36.2 TPM for the control. The UTGs in groups, C, B, and F were not preferentially up-regulated under drought stress (Figure 7B and Figure 8).

### 3.7. Co-Expression Networking of Putative UGTs

Out of the 191 identified UTGs, 71 UGT identifiers were found in the famNet co-expression network database. Only 12 UGT genes were found to be co-expressed with their closest neighboring genes (Appendix A). The remaining UGTs were independently expressed. These 12 UGTs were found to be co-expressed with cytochrome P450, prenyl-transferases, plant lipid transfer proteins, hemolysin-III, and protein-tyrosine phosphatase-like genes (Appendix A). For example, *Potri.016G014500* (*UGT71*) from group E, *Potri.006G179700* (*UGT79*) from group A, *Potri.016G097400* (*UGT73*) from group D, and *Potri.009G133300* (*UGT78*) from group F were found to be co-expressed with cytochrome P450 genes as their closest neighboring gene. *Potri.016G022500* (*UGT85*) from group G was found to be co-expressed with both prenyl-transferases and tyrosine phosphatase-like genes, and *Potri.003G210400* (UGT92) from group M was found to be co-expressed with a glucose-methanol-choline oxidoreductase gene. Some genes showed duplicated modules within *Populus* (Appendix A).

## 4. Discussion

Plant UGTs are known to be associated with a number of physiological functions, including seed, flower, fiber, root, stigma, and fruit development; delayed senescence; salt and oxidative stress tolerance; response to UV-B radiation and drought stress; mycotoxin inactivation; and fungal resistance [62]. They are the strongest candidates for hormonal regulation in plants by virtue of their role in catalyzing the formation of glycoconjugates of myo-inositol esters of IAA, OxIAA-glucoside, ABA-glucoside, IBA conjugates, ABA glucose ester, 1-*O*-indole acetyl glucose ester, brassinolide-23-*O*-glucosideBR malonyl glucosides, trans-zeatin *O*-glucosides, cytokinin *N*-glucosides, tuberonic acid glucoside, and salicylic acid glucoside [62]. The UGT multigene family has been identified in several plant species, including eudicots and monocots, including *P. persica*, *M. domestica*, *V. vinifera*, *L. usitatissimum*, *G. max*, *Z. mays*, *G. soja*, *G. hirsutum*, *Eucommia ulmoides* [12,18,19,20,21,22,23,24,25,26,27,28]. *Populus* is considered as a model plant for forest tree research; however, only two UGTs have been characterized and found to be associated with flavonoid glycosylation in *Populus* [5]. Thus, it is essential to extend the study of UGTs in *Populus* to determine their roles in a diversity of plant physiological functions.

In this study, we identified 191 UGT genes in *Populus* that code for 0.3% of the total transcripts present in the *Populus* genome. Similar percentages of transcriptomes comprising UGT genes have been observed in *L. usitatissimum* (0.31%), *M. domestica* (0.39%), and *P. persica* (0.35%) [12,18,25]. In contrast, *Arabidopsis* (0.22%), *G. raimondii* (0.16%) and *G. max* (0.16%) all have lower proportions of the transcriptome comprising UGT genes compared with *Populus* [17,20,23]. Among the examined plants, *V. vinifera* has the highest percentage of the transcriptome (0.69%) comprising UGT genes [12]. These results confirm that the UGT family is widely distributed among vascular plants [12].

Structural investigation of the PSPG motifs in each phylogenetic group revealed the role of specific amino acid residues that are highly conserved at positions 1 (W), 4 (Q), 10 (H), 19 (H), and 44 (Q) (Figure 1B). The occurrence of these specific amino acids at these positions in the sequence provides certain evolutionary and functional information that could be helpful for enzyme discovery [12].

On the basis of phylogenetic analysis, all the 191 UGTs were clustered into 16 phylogenetic groups (A–P), including two newly defined groups, O and P (Figure 1A). Groups O and P groups are observed in all the examined eudicots, including *Populus*, although they are absent in *Arabidopsis,* indicating that they have been lost at some stage during the evolution of this plant (Figure 1A) [12]. Only one member of group N was detected in dicot plants, whereas the same group is larger in monocots [25]. In addition, *Populus* has a larger number of UGT genes than *Arabidopsis* (107 members), which is due primarily to an expansion within groups E, D, G, and M. This expansion indicates that in *Populus* these groups of UGTs are involved in several secondary metabolite glycosylations.

Groups E, D, G, L, and M were the most expanded of the identified phylogenetic groups, indicating that multiple functions are associated with these groups of UGTs, and they have a broad substrate specificity. Groups B, C, and O are not expanded in eudicots, which suggests that they have a conserved substrate specificity [12] (Figure 2). Surprisingly, among all eudicots, group H is the only phylogenetic group with a lower number of UGT genes compared with *Arabidopsis*, which indicates that these UGTs are no longer required in eudicots and have a limited function associated with them. An expansion, conservation, and reduction of UGT genes in each phylogenetic group of eudicots reflects the physiological challenges that plant has to overcome for survival on land [12].

To understand the evolution of UGTs, a knowledge of intron gain and loss events and the positions and phases of introns relative to the protein sequence is very important [25]. An equal percentage (43.45%) of intron-less UGT genes are found in *Populus* and *P. persica*, which is close to the percentage for flax (40%) and peach (42%), but lesser than that for *Z. mays* (60%) and *Arabidopsis* (58%) [17,18,21,25]. Conserved intron is the most widespread and oldest intron observed in most members of the different phylogenetic groups in *Populus*. In *Z. mays*, *P. persica,* and flax, intron 5 is also considered to be the oldest intron [18,21,25] (Figure 5). Most of the intron insertions were observed in phase 1 across all the compared eudicots gene structures, suggesting that the majority of conserved introns are ancient elements and that their phases remain stable [25,63] (Appendix A). Intron size within each phylogenetic group appears to be variable (Appendix A), suggesting that intron size was gene-specific during evolution [23].

With the exception of chromosome 20, the 191 *Populus* UGTs are dispersed throughout all the chromosomes of the *Populus* genome (Figure 3). A clustering of 2 to 15 genes per cluster showed high-sequence similarity and were frequently classified into the same phylogenetic group. This tendency reflects the occurrence of recent gene duplication events and close phylogenetic relationships, which is consistent with the findings for *Arabidopsis,* soybean, and cotton UGTs [20,23,31,64]. Gene duplication plays a critical role in increasing the number of these genes, the generation of new genes, and dispersing them throughout the genome [59]. The expansion of the UGT gene family is primarily due to WGD events (Appendix A). Within the *Populus* genome, 65 UGTs have been segmentally duplicated and three UGTs have been tandemly duplicated, suggesting that segmental duplication has been the dominant form of duplication during the evolution of UGTs in *Populus* (Appendix A) [23]. These results are consistent with data showing that the *Populus* genome has undergone at least one round of WGD [4]. Some UGTs from groups D and L have evolved through a series of duplications, indicating that they have been individually duplicated and later became the part of the genome. Sixty-one percent of the segmentally duplicated UGTs were duplicated around 11–65 mya, which indicates that the genome duplication in *Populus* is very recent 8 to 13 mya, as reported by [64]. However, the fossil record shows that the *Populus* and *Salix* lineages diverged 60 to 65 mya [4]. All the three tandemly duplicated UGT genes were duplicated 19–65 mya, suggesting that both tandem and segmental duplication events occurred at the same time as WGD (Appendix A).

Prior to the current study, the role of UGTs in drought tolerance was not clearly defined. The exception being the *UGT74E2* and *UGT76E11* genes in *Arabidopsis*, which has been found to be involved in IBA glycosylation and flavonoid accumulation which is associated with the water stress response [39,65]. A high expression of all UGTs across leaves and roots suggests their drought responsiveness because these are the primary tissues affected under water deficit conditions. Eighty-four droughts responsive UGTs from *Populus* were marked based on microarray and RNA-seq data (Figure 7A, B). The UGTs in phylogenetic groups E, B, C, D, K, F, and L were down-regulated in *Populus*, showing differential expression between control and drought treatments (Figure 7A). Most of the down-regulated UGTs from phylogenetic group E belong to subgroup UGT71 genes, which have been functionally characterized for ABA glycosylation in strawberry and *Arabidopsis* [43,44,66,67]. Genes in the other two subgroups, *UGT72* and *UGT88*, from phylogenetic group E are also down regulated but their functional characterization regarding hormonal glycosylation has yet to be reported (Figure 7A). Hence, they could be considered target genes to investigate their substrate specificity for drought responsive phytohormones. The UGT gene *Potri.016G014500* (*UGT71*) from phylogenetic group E was the most down-regulated UGT observed under drought conditions. All the subgroups UGT74, UGT75, and UGT84 were found to be negatively regulated under drought conditions and have already been characterized for glycosylation of IAA and IBA in *Arabidopsis* [39,45,68,69]. In *Arabidopsis*, *UGT84A1*, *A2*, *A3*, and *A4* have been functionally characterized against UV-B radiation by [70]. In *G. hirsutum*, *Arabidopsis*, and tomato, *UGT73* from group D has been characterized for glycosylation of ABA, brassinosteroids, and salicylic acid [42,50,66,67,68]. The UGTs of groups B, C, K, and F still need to be assessed for their putative role in drought response.

Phylogenetic groups E, A, M, D, O, L, I, J, H, P, and G were actively up-regulated in response to drought conditions (Figure 7B). Again, the most up-regulated UGTs belong to groups E and L, indicating that these UTGs are both up- and down-regulated under drought stress. Regarding the UGTs from group J, *UGT87A2* from *Arabidopsis* has been characterized for ABA glycosylation (P. Li et al., 2017). We propose here the UGTs from phylogenetic groups M, O, I, H, and P are potential candidates for hormonal substrate specificity, and still need to be assessed regarding their response to drought. Overall, we obtained consistent results based on the microarray data and RNA-seq data generated in this study.

Most of the UGTs in *Populus* were found to be expressed independently under specific circumstances. The maximum co-expression of UGTs with cytochrome P450 from phylogenetic groups E, A, D, and F confirmed that they are members of a triterpenoid biosynthesis pathway involved in several secondary metabolite glycosylations (Appendix A).

## 5. Conclusions

We identified 191 UGT genes in the *Populus* genome. On the basis of phylogenetic analysis, these genes were found to cluster into 16 distinct evolutionary groups (A–P). The PSPG motif amino acid sequence was found to vary within each phylogenetic group, with the exception of amino acid positions 1, 4, 10, 19, and 44. An intron between amino acids 150 and 200 is the oldest intron found across eudicots, including *Populus*. Segmental duplication in the *Populus* genome contributed dominantly during the evolution of the UGT family. Most of the identified UGT genes are expressed in roots, leaves, flowers, and seeds. The results of digital expression analysis were confirmed by RNA-seq data. Eighty-four UGT genes were found to be up- and down-regulated in response to drought, and were shown to be putatively involved in ABA, IBA, and IAA glycosylation. Phytohormonal glycosylation is the most important phenomena associated with UGTs in their response to drought stress. Molecular evolution and transcriptome analyses can be useful for understanding the structure–function relatedness of the UGT family members and might further facilitate their functional analysis.

## Figures and Tables

**Figure 1 genes-13-01640-f001:**
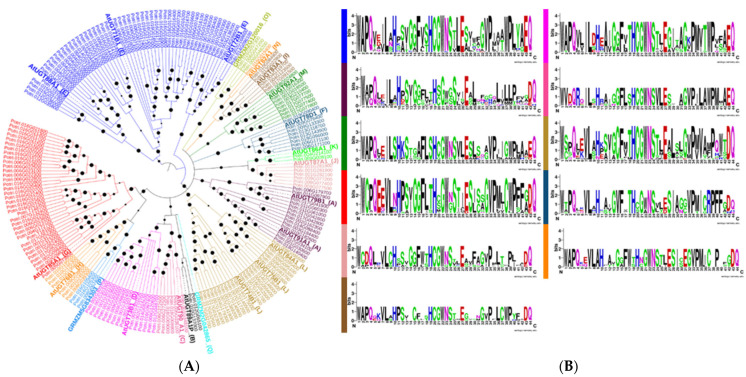
**Phylogenetic analysis of 191 *Populus* UDP glycosyltransferases (UGTs) and their PSPG motifs in each phylogenetic group (A,B)**. (**A**) Sixteen major phylogenetic groups (A–P) were found in *Populus*. The evolutionary history was inferred using the Neighbor-joining method. The optimal tree with the sum of branch lengths = 14,369.13429772 is shown. The tree is drawn to scale, with branch lengths in the same units as those of the evolutionary distances used to infer the phylogenetic tree. The evolutionary distances were computed using the number of differences method and are in the units of the number of amino acid differences per sequence. The analysis involved 213 amino acid sequences. All ambiguous positions were removed for each sequence pair. There were a total of 503 positions in the final dataset. Evolutionary analyses were conducted in MEGA7. Bootstrap values are shown in black dots at each node. (**B**) Web logos of PSPG motifs from different phylogenetic groups are shown in different colors.

**Figure 2 genes-13-01640-f002:**
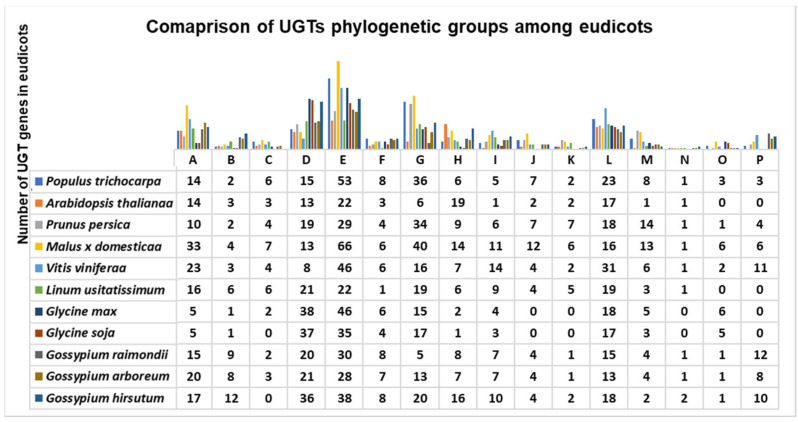
**Comparison of *Populus* UGT phylogenetic groups with those of other published eudicots**. Sixteen major phylogenetic groups (A–P) can be observed in all putative UGTs from eudicots. Phylogenetic groups such as D, E, G, L and M were expanded in all eudicots while the group N was the most conserved phylogenetic group observed in all eudicots.

**Figure 3 genes-13-01640-f003:**
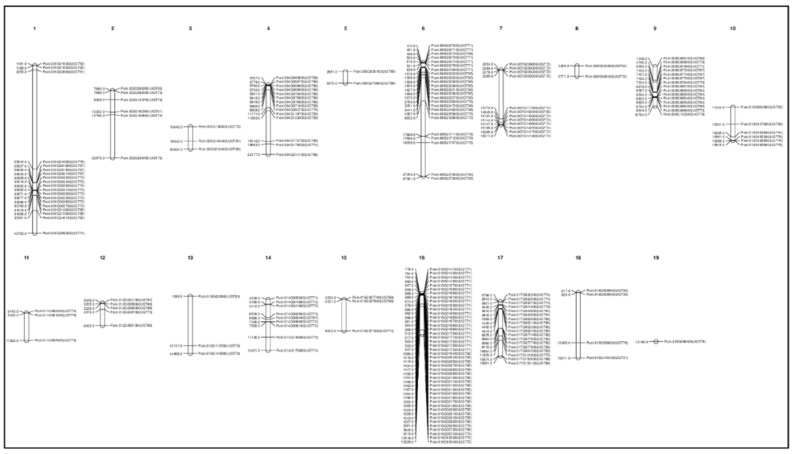
**Chromosomal distribution of UDP glycosyltransferases (UGTs) across the *Populus* genome against the type of UGT gene**. Except chromosome 04, all the remaining 19 chromosomes had UGT genes. The maximum number of UGT genes were found on chromosome 16.

**Figure 4 genes-13-01640-f004:**
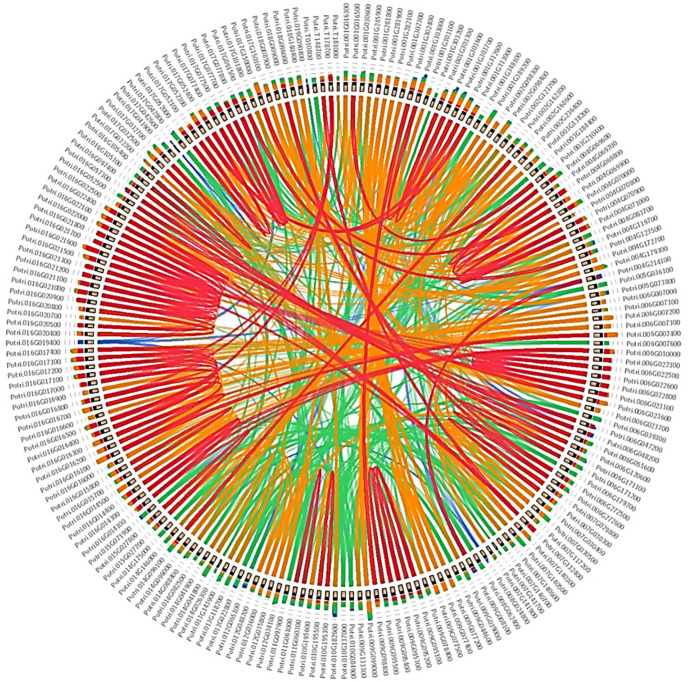
**Syntenic analysis of 191 UDP glycosyltransferases (UGTs) in *Populus***. Different color threads show the duplication of genes among all UGTs presented.

**Figure 5 genes-13-01640-f005:**
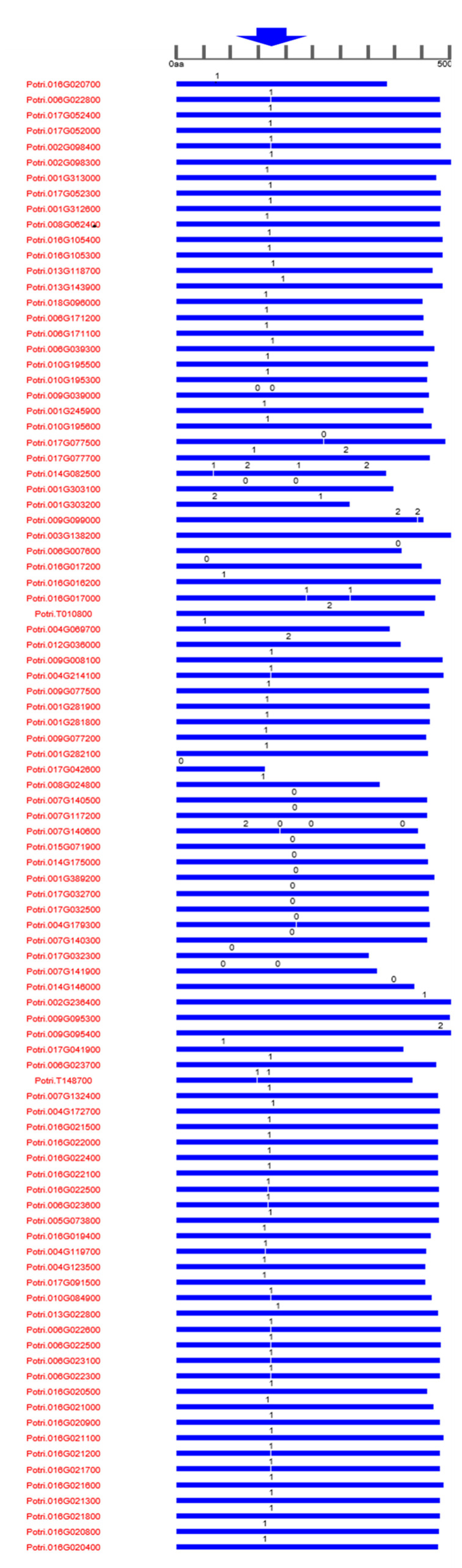
**Intron organization of all intron containing UDP glycosyltransferases (UGTs) in *Populus***. Introns between 150–200 amino acids were found maximally conserved regarding their insertion and phase.

**Figure 6 genes-13-01640-f006:**
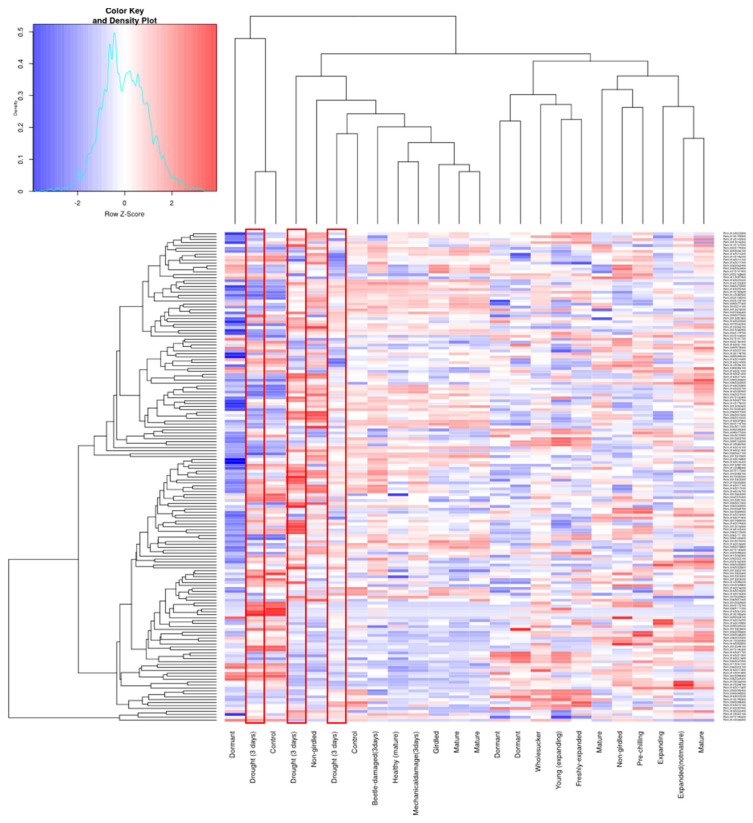
**Heat map of 168 expressed UDP glycosyltransferases (UGTs) across different treatments in the *Populus* genome**. The highlighted areas show the up- and down-regulated UGT genes against drought treatment.

**Figure 7 genes-13-01640-f007:**
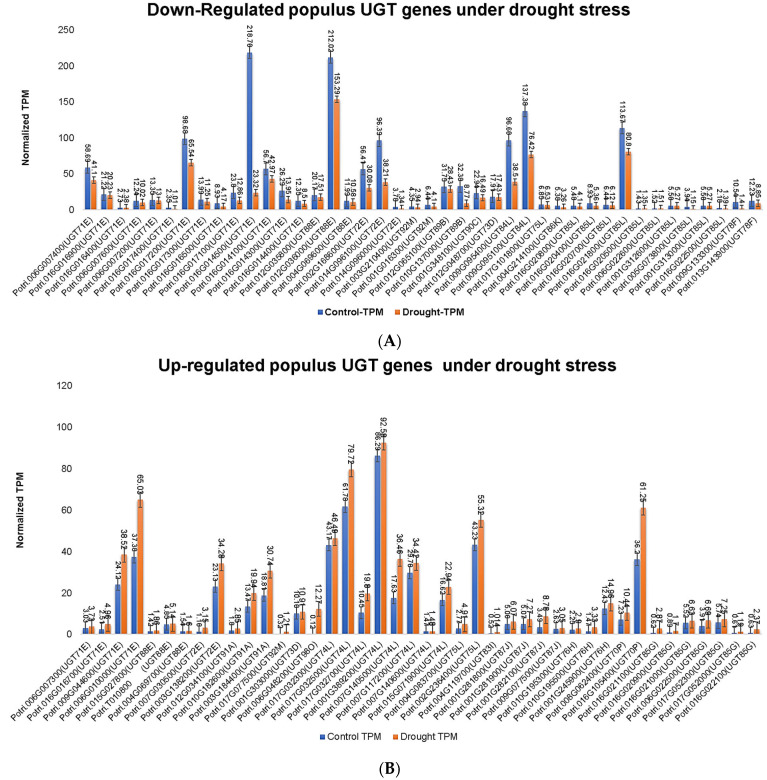
**Regulation of UDP glycosyltransferases (UGTs) under drought conditions with their gene expression values in TPM (A,B)**. (**A**) Down-regulation of UGTs under drought and control conditions. (**B**) Up-regulation of UGTs under drought and control conditions.

**Figure 8 genes-13-01640-f008:**
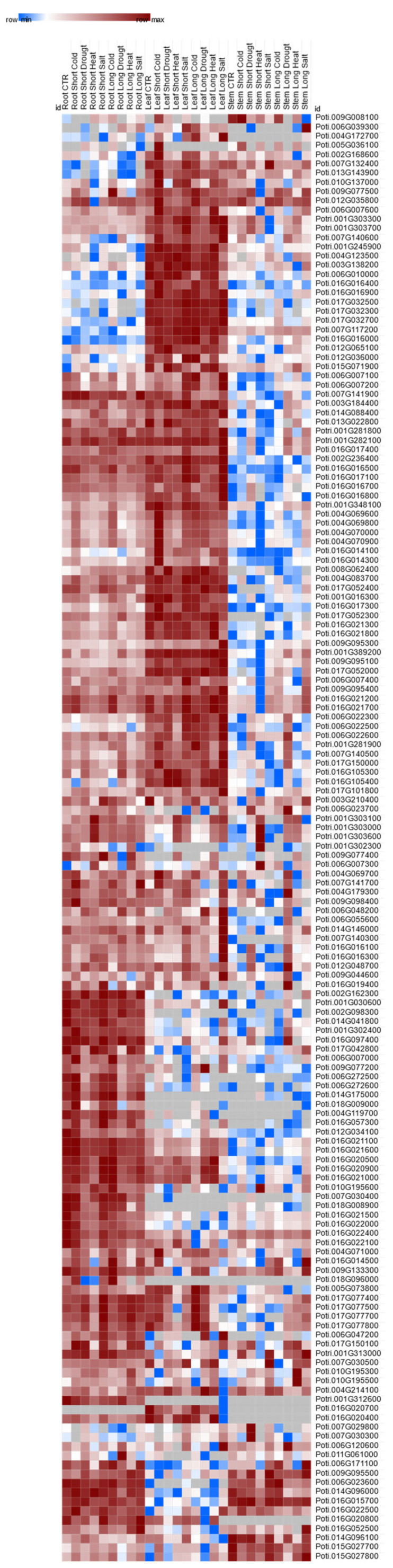
Transcriptome heatmap of UDP glycosyltransferases (UGTs) under different abiotic stress conditions.

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
