# Peer review of "Genome Wide Analysis of Family-1 UDP Glycosyltransferases in Populus trichocarpa Specifies Abiotic Stress Responsive Glycosylation Mechanisms"

_genes, 2022, doi:10.3390/genes13091640_

Round 1
Reviewer 1 Report
The research (Genome-wide analysis of family-1 UDP glycosyltransferases in 2 Populus trichocarpa specifies abiotic stress-responsive glycosyl-3tion mechanisms) employed bioinformatics techniques to identify 191 putative UGTs in Populus (a model species for perennial woody plants), and confirmed the transcriptomic data on UGT expression in plant tissues and under drought stress. They also designed UGT expression models for up- and down-regulation of UGT genes under drought and control conditions.
The manuscript idea is interesting but there are some minor revisions:
1. Figure 6 and figure 8 have a very weak resolution.
2. The manuscript needs English language and grammar revision.
Author Response
Thank you very much for your comments to improve our manuscript.
Q#1Figure 6 and figure 8 have a very weak resolution.
Ans#1 In this figure, We have shown the overall expression patterns of all 191 UGTs in Popoulus, The highlighted areas such as drought based expression is shown seprately in Figure 7.
Q#2 The manuscript needs English language and grammar revision
Ans#2 Before we submitted this manuscript, We have already revised by English editors to improve it.
Reviewer 2 Report
It is necessary to provide a more complete description of the experiment in the Materials and Methods section, for example, there are no reports of sampling.
There is not enough visualization of the material, the use of histochemical methods would beautify the work.
Figure 8 in the Discussion section is hardly needed.
Figures 1, 3, 5, 6, 8 are poorly readable, it is necessary to increase the font.
It is necessary to issue a list of references according to the requirements of the journal.
Author Response
Q#1 It is necessary to provide a more complete description of the experiment in the Materials and Methods section, for example, there are no reports of sampling.
Ans#1 Regarding material and methods we employ various bioinformatic tools to elucidate our manuscript and all methods have been described briefly.
There is not enough visualization of the material, the use of histochemical methods would beautify the work.
Ans#2 We did not employ any wetlab lab experiment hence we can not show histochemical results.
Figure 8 in the Discussion section is hardly needed.
Ans# 8 It shows an earlier expression under various stresses as well such as cold, heat and salt hence we want to extend our UGTs expression patterns in other stresses.
Figures 1, 3, 5, 6, 8 are poorly readable, it is necessary to increase the font.
A wholistic view in all figures minimize its resolution. hence we provided all the important data in supplementary tables.
It is necessary to issue a list of references according to the requirements of the journal.
The references has been set according to the journal format.
Reviewer 3 Report
In the current manuscript, Rehman et al., employed bioinformatics techniques to identify putative UGTs in Populus, and then subjected these to genomic and expression analyses. They then further confirmed the publicly available microarray and transcriptomic data on UGT expression in plant tissues and under drought stress. Also, they designed UGT expression models for up- and down-regulation of UGT genes under drought and control conditions. Finally, co-expression networks were constructed in order to gain a better understanding of the involvement of these genes in other pathways. Although the topic is attractive, there are some concerns that should be addressed.
- Please follow the format style of the journal for citing references
-There are some typographical and grammatical errors.
- Discussion should be improved.
- The conclusion section is very short. At least it should discuss more future work
Author Response
Q#1 Please follow the format style of the journal for citing references
Ans#1 it is accordingly now.
-There are some typographical and grammatical errors.
Ans#2 All the errors were removed
Q#3 Discussion should be improved.
We have already cited enough references to support out results. Much experimental data about UGTs has not been reported yet.
- The conclusion section is very short. At least it should discuss more future work
We believe that over emphasize our conclusion can be misleading hence we keep it simple but clear.